# Astragalin and Isoquercitrin Isolated from *Aster scaber* Suppress LPS-Induced Neuroinflammatory Responses in Microglia and Mice

**DOI:** 10.3390/foods11101505

**Published:** 2022-05-22

**Authors:** Eun Hae Kim, Youn Young Shim, Hye In Lee, Sanghyun Lee, Martin J. T. Reaney, Mi Ja Chung

**Affiliations:** 1Department of Food Science and Nutrition, College of Health Welfare, Gwangju University, Gwangju 61743, Korea; kjr01056@naver.com (E.H.K.); lhi1156@hanmail.net (H.I.L.); 2Department of Plant Sciences, University of Saskatchewan, Saskatoon, SK S7N 5A8, Canada; younyoung.shim@usask.ca (Y.Y.S.); martin.reaney@usask.ca (M.J.T.R.); 3Prairie Tide Diversified Inc., Saskatoon, SK S7J 0R1, Canada; 4Department of Integrative Biotechnology, Biomedical Institute for Convergence at SKKU (BICS), Sungkyunkwan University, Suwon 16419, Korea; 5Department of Plant Science and Technology, Chung-Ang University, Anseong 17546, Korea; slee@cau.ac.kr; 6Saskatchewan Oilseed Joint Laboratory, Department of Food Science and Engineering, Jinan University, Guangzhou 510632, China

**Keywords:** cytokines, microglia, hippocampus, neuroinflammation, *Aster scaber*, nitric oxide

## Abstract

The current study investigated the anti-neuroinflammatory effects and mechanisms of astragalin (Ast) and isoquercitrin (Que) isolated from chamchwi (*Aster scaber* Thunb.) in the lipopolysaccharide (LPS)-activated microglia and hippocampus of LPS induced mice. LPS induced increased cytotoxicity, nitric oxide (NO) production, antioxidant activity, reactive oxygen species (ROS), inducible nitric oxide synthase (iNOS) expression, the release of pro-inflammatory cytokines, protein kinase B phosphorylation, and mitogen-activated protein kinases (MAPK) phosphorylation in LPS-treated microglial cells. Intraperitoneal injection of LPS also induced neuroinflammatory effects in the murine hippocampus. Ast and Que significantly reduced LPS-induced production of NO, iNOS, and pro-inflammatory cytokines in the microglia and hippocampus of mice. Therefore, anti-inflammatory effects on MAPK signaling pathways mediate microglial cell and hippocampus inflammation. In LPS-activated microglia and hippocampus of LPS-induced mice, Ast or Que inhibited MAPK kinase phosphorylation by extracellular signal-regulated kinase, c-Jun N-terminal kinase, and p38 signaling proteins. Ast and Que inhibited LPS-induced ROS generation in microglia and increased 1,1-diphenyl-2-picrylhydrazyl radical scavenging. In addition, LPS treatment increased the heme oxygenase-1 level, which was further elevated after Ast or Que treatments. Ast and Que exert anti-neuroinflammatory activity by down-regulation of MAPKs signaling pathways in LPS-activated microglia and hippocampus of mice.

## 1. Introduction

The central nervous system (CNS) is inaccessible to cells of the systemic immune system due to the blood–brain barrier. The principal immune cells in the CNS are the microglial cells, which make up 10–12% of brain cell populations [1,2]. The highest density of microglia is found in the hippocampus [3]. Microglia can respond to brain tissue injury if activated by typical pathological conditions. Increases in the production of pro-inflammatory cytokines (interleukin-1β (IL-1β), IL-6, and tumor necrosis factor-α (TNF-α)) and pro-inflammatory enzymes (inducible nitric oxide synthase (iNOS) and cyclooxygenase-2) [4] observed in microglial can have been associated with progressive neuronal cell death. Neurodegenerative diseases, such as Alzheimer’s disease (AD) and Parkinson’s disease (PD) [5,6,7], occur as neuronal death becomes evident. The symptoms of AD dementia include cognitive and memory impairment. AD is characterized by microglia-mediated neuroinflammation. Therefore, therapeutic agents for treating AD might be identified through their potential to mitigate neuroinflammation.

AD pathogenesis has been associated with oxidative stress. Microglia respond to activation through the synthesis of intracellular reactive oxygen species (ROS). Furthermore, activation can lead to the expression of mitogen-activated protein kinases (MAPK) signaling pathways, including extracellular signal regulated kinase (Erk), c-Jun N-terminal kinase (JNK), and p38 [8]. Potential anti-inflammatory therapeutics are being investigated for their ability to increase MAPK activity and regulate the transcription and translation of inflammation mediators. Preclinical data indicate that inhibitors targeting the p38 and JNK pathways induce anti-inflammatory effects [9,10]. Neuroinflammation in lipopolysaccharide (LPS)-activated microglia is mediated by expression of the phosphatidylinositol 3-kinase (P13K)/protein kinase B (Akt) signaling pathway [11,12].

The expression of pro-inflammatory cytokines is reduced by extracts rich in pure flavonoids (e.g., quercitrin, genistein, astragalin (Ast or A), isoquercitrin (Que or Q)) and polyphenols. These compounds further reduce, as well as down-regulate MAPKs and P13K/Akt signaling pathways [9,12,13]. Heme Oxygenase-1 (HO-1) is an inducible antioxidant thought to be important for cells against the oxidative stress and may contribute to cell protection against ROS-induced cell death [14,15]. Study results suggest that the anti-inflammatory effect of quercetin in the BV-2 microglial cell lines is associated with the induction of HO-1 [10]. Therefore, modulation of HO-1 could be a therapeutic target to attenuate neurodegenerative diseases [16].

Chamchwi *(**Aster scaber* Thunb.), an edible plant rich in flavonoids, is widely cultivated as a culinary vegetable in Korea [17]. The rich nutrients contained in *A. scaber* are vitamin C, Ca, Fe, and β-carotene [18]. *A. Scaber* leaves contain caffeoylquinic acid compounds, flavonoids, and terpenoids [19,20]. *A. Scaber* had the potential antioxidant and anti-obesity effects [21] due to radical scavenging-linked anti-adipogenic activity [20] and has protective effects against oxidative stress-induced human brain cell death [17]. In our previous study, Ast (kaempferol-3-*O*-glucoside) and Que (quercetin-3-*O*-glucoside) were isolated from *A. scaber*. Ast treatment of human brain neuroblastoma SK-N-SH cells exposed to H_2_O_2_ down-regulated the MAPK pathway and up-regulated HO-1 [17]. Ast down-regulates the NF-kB signaling pathway [22] and thereby induces numerous responses, including anti-allergic effects [23], cardioprotective effects [24], anti-cancer effects [25], and anti-inflammatory effects [26]. Que induces anti-inflammatory responses through inhibition of the NF-*k*B/MAPKs signaling pathway [27,28]. Que bioactivity includes antioxidant effects [29], hepatoprotective effects [30], antiviral activities [31], and a neuroprotective effect on Parkinson’s disease [32]. However, there are no reports describing the anti-neuroinflammatory effect of Ast and Que *in vivo* and *in vitro* through down-regulation of the NF-*k*B/MAPKs signaling pathway.

From both *in vitro* microglia and *in vivo* animal models, it has been shown that LPS induces neuroinflammation by increasing inflammatory mediators [33]. In this study, Ast and Que’s anti-neuroinflammatory effects and action mechanism isolated from *A. scaber* were investigated in LPS-activated microglia and hippocampus of LPS induced mice. Therefore, we investigated Ast and Que as functional materials for improving neurodegenerative disorders, such as AD-related to neuroinflammation.

## 2. Materials and Methods

### 2.1. Plant Materials

Edible Korean chamchwi, *A. scaber* Thunb. (also known as *Doellingeria scabra* Thunb.) was collected in June 2017 from Yangyang (Korea). The specimen was authenticated by Korea National Arboretum (Pocheon, Korea).

### 2.2. Reagents and Apparatus

Trypsin-ethylenediaminetetraacetic acid (EDTA), fetal bovine serum (FBS), Dulbecco’s phosphate-buffered saline (D-PBS), and mixed antibiotic (penicillin/streptomycin) were purchased from Gibco-BRL (Grand Island, NY, USA). Ascorbic acid, 3-(4,5-dimethylthiazol-2-yl)-2,5-diphenyl tetrazolium bromide (MTT), 2,7-dichlorofluorescein diacetate (DCF-DA), dimethyl sulfoxide (DMSO), and 1,1-diphenyl-2-picrylhydrazyl (DPPH) were purchased from Sigma-Aldrich Co. (St. Louis, MO, USA). The culture medium (Dulbecco’s modified Eagle’s medium or DMEM) was purchased from the American Type Culture Collection (ATCC, Manassas, VA, USA). The HO antibody was obtained from Santa Cruz Biotechnology (Santa Cruz, CA, USA). The phospho-Akt, Akt, phospho-Erk, Erk, phospho-p38, p-38, phospho-JNK, JNK, iNOS, β-actin, and horseradish peroxidase (HRP)-conjugated anti-rabbit IgG antibody, and HRP-conjugated anti-mouse IgG antibody were obtained from Cell Signaling Technology (Beverly, MA, USA). Ethanol (EtOH), methanol (MeOH), and dichloromethane were purchased from SamChun Pure Chemical Co. (Pyeongtaek, Korea).

### 2.3. Isolation of Phytochemical Constituents from A. scaber

Dried and ground *A. scaber* was extracted with 10 volumes of 70% EtOH at 80 °C for 8 h three times, and the ethyl acetate (EtOAc) fraction from 70% EtOH extract was obtained [17]. The 70% ethanol extract of *A. scaber* was filtered, concentrated, and lyophilized, and the abbreviation of lyophilized sample is AE. Ast and Que (Figure 1B) were isolated from the EtOAc fraction of *A. scaber* [17].

### 2.4. Microglial Cells Culture

The EOC-20 mouse microglia was purchased from the ATCC. The microglial cells were kept in DMEM containing 10% FBS and 1% penicillin/streptomycin in a humidified incubator (Thermo Fisher Scientific, Waltham, MA, USA) with 5% CO_2_ at 37 °C. Cell cultures were inoculated at a density of 1 × 10^4^ cells per well in 24- or 96-well plastic culture plates. Subsequently, cultured cells were assayed for viability (MTT assay) and intracellular ROS. For reverse transcription-polymerase chain reaction (RT-PCR) and Western blot analysis, cultures were inoculated at a density of 1 × 10^6^ cells per well in 6-well plastic culture plates then grown until 60–80% confluent.

### 2.5. Cell Viability and NO Content

Cell viability was determined by the MTT method based on mitochondrial dehydrogenase activity [17]. After Ast or Que treatments of samples for 24 h, the cells were incubated for 4 h with fresh media containing MTT (final concentration; 0.5 mg/mL). Living cells convert MTT to formazan; after formazan formation, the media with residual MTT was removed, and the formazan was extracted with DMSO. The solution absorbance at 560–570 nm was measured using a microplate reader (Molecular Devices Co., Sunnyvale, CA, USA). Cell viability was expressed as the percentage of viable cells in the LPS and sample-free control group. The microglial cells were co-incubated with DMEM supplemented with Ast or Que samples and LPS (0.5 μg/mL) together for 12 h. the Griess reagent assay was employed to measure NO and its oxidative metabolite, nitrite [34]. In brief, in a 1.5-mL tube, 300 μL of culture supernatants were mixed with an equal volume of Griess reagent (1:1 mixture (*v*/*v*) of 1% sulfanilamide in 5% phosphoric acid and 0.1% *N*-(1-naphthyl)-ethylenediamine in water). The reagent and supernatant were mixed and then maintained in the dark at room temperature for 5 min. Sample nitrite content was determined using absorbance at 540 nm and compared with a standard curve prepared from fresh sodium nitrite.

### 2.6. Animal Treatments

Male C57BL/6 mice, aged 8 weeks, were purchased from Orient Bio Inc. (Seongnami, Korea) one week prior to the experiment and were housed in a room with controlled air temperature (21–25 °C) and humidity. The day/night cycle was 12 h on/12 h off. Mice were divided into 8 groups, and the number of mice in each group was 6–8. Animal care and handling were carried out in accordance with the protocols approved by the Committee on Animal Experimentation of the World Institute of Kimchi (permit number: WIKIM IACUC 201812). A schematic diagram of the treatment schedule is shown in Figure 2.

The mice were randomly divided into 8 groups: (1) Control group; (2) LPS group (intraperitioneal (i.p.) 300 μg/kg body weight (bw) = 7.5 μg LPS in 0.2 mL/mouse (25 g)); (3) A5 (i.p. 5 mg/kg bw = 125 μg Ast in 0.2 mL/mouse) → LPS (i.p. 300 μg/kg); (4) A20 (i.p. 20 mg/kg bw = 500 μg Ast in 0.2 mL/mouse) → LPS (i.p. 300 μg/kg); (5) Q1 (i.p. 1 mg/kg bw = 25 μg Que in 0.2 mL/mouse) → LPS (i.p. 300 μg/kg); (6) Q5 (i.p. 5 mg/kg bw = 125 μg Que in 0.2 mL/mouse) → LPS (i.p. 300 μg/kg); (7) AE5 (i.p. 5 mg/kg bw = 125 μg AE in 0.2 mL/mouse) → LPS (i.p. 300 μg/kg); and (8) AE20 (i.p. 20 mg/kg bw = 500 μg AE in 0.2 mL/mouse) → LPS (i.p. 300 μg/kg). Inflammatory progression was induced in each mouse by i.p. injection of LPS (5 mg/kg) for 4 days. The Ast, Que or AE were injected intraperitoneally every day before i.p. injection of LPS (i.p. 300 μg/kg). The control group was exposed to DPBS by i.p. injection without Ast, Que and AE. LPS was dissolved in D-PBS. Ast, Que, and AE were dissolved in DMSO and then diluted with D-PBS.

After feeding at 17:00, the mice were fasted for 16–19 h (overnight) and sacrificed on day 5. The mice were killed by carbon dioxide (CO_2_) inhalation using a chamber filled with CO_2_. The hippocampus of the brain was carefully removed for further study. The hippocampus was divided into three parts, and the hippocampus was rapidly frozen in liquid nitrogen for RT-PCR, ELISA and Western blot analysis.

### 2.7. Total RNA Extraction and RT-PCR

The microglial cells were treated with Ast or Que samples together with LPS (0.5 μg/mL) for 12 h, and the hippocampus tissues were prepared after the animal experiment. Total RNA was extracted and isolated from cells using the Easy-spin™ total RNA extraction kit (iNtRON Biotechnology, Seongnam, Korea) according to the manufacturer’s protocol. The RNA was reverse transcribed using a Power cDNA synthesis kit (iNtRON Biotechnology, Seongnam, Korea) based on the manufacturer’s guidelines. PCR was performed after adding forward (F) and reverse (R) primers (20 μL of each primer 10 pmol/μL) to a PCR premix kit (Maxime™, iNtRON Biotechnology, Seongnam, Korea). The following primers were used: iNOS, F, 5′-CTG CAG CAC TTG GAT CAG GAA CCT G-3′ and R, 5′-GGG AGT AGC CTG TGT GCA CCT GGA A-3′; COX-2, F, 5′-GGA GAG ACT ATC AAG ATA GTG ATC-3′ and R, 5′-ATG GTC AGT AGA CTT TTA CGA CTA -3′; TNF-α, F, 5′-CCC CTC AGC AAA CCA CCA AGT-3′ and R, 5′-CTT GGG CAG ATT GAC CTC AGC-3′; β-action, F, 5′-TGG AAT CCT GTG GCA TCC ATG AAA C-3′ and R, 5′-TAA AAC GCA GCT CAG TAA CAG TCC G-3′. The PCRs with the iNOS, COX-2, TNF-α, and β-actin primers were performed using an initial cycle at 94 °C for 5 min. This was followed by 30 cycles (iNOS, COX-2, TNF-α) or 25 cycles (β-action) at 94 °C for 20 s, 55 °C for 20 s, and 72 °C for 1 min, with a final extension at 72°C for 7 min. The β-action transcripts were used as an internal control when PCR was performed. In addition, PCR products were separated by electrophoresis on a 2% agarose gel, and bands were quantified using the SigmaGel 1.0 (Jandel Scientific, San Rafael, CA, USA).

### 2.8. Western Blot Analysis

After pretreatment with Ast or Que in DMEM medium for 2 h microglial cells were treated with DMEM with LPS (0.5 μg/mL) for 30 min. But the microglial cells were treated with Ast or Que samples and 0.5 μg/mL LPS for 12 h for iNOS and HO-1 protein analysis. Cells were lysed in RIPA lysis buffer (50 mM Tris, 150 mM NaCl, 2 mM EDTA, 1% Triton X-100, 0.1% SDS, pH 7.8) containing protease and phosphatase inhibitor cocktails (Roche Applied Science, Mannheim, Germany). Lysates were then centrifuged (14,000 rpm, 4 °C, 15 min). Frozen (liquid N_2_) hippocampus tissue was thawed and then ground after adding RIPA buffer. After centrifugation (14,000 rpm, 4 °C, for 15 min), the supernatants were used for Western blot analysis. The protein concentration of the lysate was measured using a Bio-Rad protein kit (Hercules, CA, USA) and compared with a bovine serum albumin (BSA, Sigma-Aldrich Co., St. Louis, MO, USA) standard calibration curve. Lanes of SDS-PAGE gels (12%) were loaded with 20 μg of lysate protein. Proteins bands in the gels were transferred to nitrocellulose blotting membranes (GE Healthcare Life Science, Freiburg, Germany). The primary antibodies (iNOS, p-Akt, Akt, p-p38, p-38, p-Erk, Erk, p-JNK, JNK, HO-1, p-NF-kB, NF-kB, and β-actin) were diluted at 1:1000 in phosphate-buffered saline (PBS) containing 0.01% Tween 20 (Sigma-Aldrich Co., St. Louis, MO, USA). The membranes were incubated with the diluted primary antibodies overnight at 4 °C. Next, the membranes were washed three times with PBS containing 0.01% Tween 20. Washed membranes were incubated with antibodies (either HRP-conjugated anti-rabbit or mouse IgG antibodies) diluted at 1:2000, as the secondary antibody, for 90 min. Western blots were used to visualize the immunoreactive proteins. An AbSignal Western blotting detection reagent kit was used for detection (AbClon, Inc., Seoul, Korea). The membrane was then exposed to an X-ray film to identify labelled protein bands. The intensities of the labelled proteins were quantified by SigmaGel software (Jandel Scientific, San Rafael, CA, USA). Protein loading was determined using a β-actin antibody.

### 2.9. Assay for TNF-α, IL-1β and IL-6 Secretion

The microglial cells were treated with Ast or Que samples and 0.5 μg/mL LPS for 12 h. The culture supernatants were harvested. The hippocampus frozen in liquid nitrogen was mixed with a PBS containing a protease inhibitor cocktail (Sigma–Aldrich Co., USA), and the hippocampus tissues were homogenized at 4 °C. The homogenate was clarified by centrifugation (14,000 rpm, 4 °C, 15 min) and then normalized for equal amounts of protein determined by the Bio-Rad protein assay (Hercules, CA, USA). The concentrations of TNF-α, IL-1β, and IL-6 released from the cells treated with LPS and samples (Ast and Que) were determined, using a mouse TNF-α, IL-1β or IL-6 enzyme-linked immunosorbent assay (ELISA) kit (eBioscience Co., San Diego, CA, USA), according to the manufacturer’s instructions.

### 2.10. ROS Level Measurement

Microglial cells were treated either with Ast or Que and 0.5 μg/mL LPS for 12 h. The DCF-DA was used to monitor free radicals’ formation by cellular oxidative stress in cells [17]. After incubating of cells with a medium containing the DCF-DA (final concentration; 50 μM) for 45 min, the medium was removed, and PBS (pH 7.4) solution was added to cells in each well, washed, and discarded twice. DCF-DA fluorescence intensity was monitored at 485 nm excitation and 538 nm using a fluorescence spectrophotometer (Synergy MX, BioTek, Winooski, VT, USA).

### 2.11. DPPH Assay

Antioxidant effects were investigated using the DPPH assay described previously [35] with slight modifications. Samples (Ast, Que, or ascorbic acid) of 540 μL at various concentrations (1, 2.5, 5, 10, 30, 60, 120 μg/mL) of Ast or Que were mixed with 360 μL of DPPH radical solution (1.5 × 10^−4^ M), and the absorbance of sample and DPPH radical solution mixture was measured at 517 nm. The control was DPPH solution plus EtOH. The inhibition percentage was calculated using the following Equation (1):Radical scavenging activity (%) = [1 − *A_sample_*_(517nm)_/*A_control_*_(517nm)_] × 100 (1)
where *A_control_* is the absorbance of the control, and *A_sample_* is the absorbance of the sample (Ast, Que, and ascorbic acid).

### 2.12. Statistical Analysis

Data were expressed as mean ± standard deviation (SD), and the average values were derived from three to eight values per experiment. The data were analyzed using the SPSS package (Version 18.0, SPSS Inc., Chicago, IL, USA) and one-way analysis of variance (ANOVA). Duncan’s multiple range test was used to determine the statistical significance (*p* < 0.05) of the different treatments.

## 3. Results

### 3.1. Cytotoxicity of Ast and Que Isolated from A. scaber

The cytotoxicity of Ast and Que on the microglial cells was evaluated using an assay of MTT capacity. The microglia were incubated with different concentrations of Ast or Que (1–10 μg/mL). The 10 μg/mL Que treatment showed cytotoxicity compared with the control (Figure 3A). It had no detectable effect on microglial cell viability compared with controls (Figure 3A). In subsequent experiments, cytotoxicity was avoided by using up to 10 μg/mL Ast and 5 μg/mL Que.

### 3.2. Effects of Ast and Que on LPS-induced Microglia

Microglial cells were treated with various doses of Ast (2.5–10.0 μg/mL) or Que (2.5, 5 μg/mL) and 0.5 μg/mL LPS for 12 h. Ast and Que strongly inhibited LPS-induced NO dose-dependent, up to 10 μg/mL Ast or 5 μg/mL Que (Figure 3B).

The effects of Ast and Que on the expression of NO-producing iNOS were tested in LPS-treated microglia (Figure 4). Ast and Que inhibited LPS-induced iNOS mRNA and protein production (Figure 4). These results indicate that LPS induces iNOS expression to induce NO production and that Ast and Que can inhibit the increase of NO and iNOS production in LPS-treated microglia.

### 3.3. Effects of Ast and Que on TNF-α and IL-1β Production in LPS-Treated Microglia

There was increased TNF-α and IL-1β release in cells exposed to 0.5 μg/mL LPS (*p* < 0.05) compared with the control (Figure 5). When simultaneous co-treatment with samples (Ast or Que) and LPS (0.5 μg/mL) was applied, the stimulatory effects of LPS on TNF-α and IL-1β protein levels were blocked by the flavonoids. At a concentration of 5 μg/mL, Ast and Que inhibited the TNF-α production by 6.7- and 5.9-fold lower than the LPS group (10.5-fold) (Figure 5A). Similarly, LPS-stimulated IL-1β release was also significantly inhibited in a dose-dependent manner by Ast and Que (Figure 5B).

### 3.4. Ast and Que Inhibit LPS-Induced Phosphorylation of the Akt/MAPK Pathway in Microglia

In LPS-activated microglial cells, MAPK and Akt expression both affect the release of pro-inflammatory cytokines. Here we investigate the effects of Ast and Que on the MAPK signaling pathways in LPS-activated microglia. This includes determining the phosphorylation of Erk, p38, JNK, and Akt (a marker of PI3-kinase activation). LPS induced the phosphorylation of the Akt/MAPK pathway, including p38, Erk, and JNK (Figure 6). Ast and Que’s treatments attenuated LPS-induced phosphorylation of the Akt/MAPK pathway. In microglia, the LPS-induced p-Akt was inhibited by Ast (5 and 10 μg/mL) and Que (2.5 and 5 μg/mL). In addition, Ast and Que significantly inhibited the phosphorylation of p38, Erk, and JNK (p-p38, p-Erk, and p-JNK). At 5 μg/mL, the phosphorylation inhibitory effects of Erk and JNK by Que were significantly higher than that of Ast.

### 3.5. Effects of Ast and Que on LPS-Induced Intracellular ROS Production and DPPH Radical Scavenging Activity

Upon stimulation with LPS, microglia accumulate intracellular ROS [36]. Although intracellular ROS levels are increased after LPS treatment, Ast and Que inhibit intracellular ROS generation in the LPS-activated microglia (Figure 7A). Que at 5 μg/mL was more potent than Ast in inhibiting intracellular ROS formation. The DPPH radical scavenging activities of Ast and Que are shown in Figure 7B. Que was a more effective DPPH radical scavenger than Ast. The DPPH radical scavenging activity of Que gradually increased dose-dependent, showing 83.5% DPPH radical scavenging activity at a concentration of 5 μg/mL compared to untreated control (0%).

### 3.6. Effects of Ast and Que on HO-1 Production in LPS-Treated Microglia

HO-1 is a stress-induced antioxidant enzyme with potential anti-inflammatory effects [37]. Hence, we examined the influences of Ast and Que on HO-1 induction. Western blot analysis revealed dramatically increased OH-1 expression after simultaneous Ast (10 μg/mL) and LPS (0.5 μg/mL) treatment compared to LPS treatment (Figure 8). Therefore, it is indicated that Ast up-regulates HO-1. In the same experiment, when microglia were co-treated with Que (2.5 μg/mL) and LPS, a statistically significant increase in HO-1 protein was observed compared to cells treated with LPS. However, the stimulatory effect of LPS on the HO-1 protein level was inhibited by adding 5 μg/mL Que.

### 3.7. Effects of Ast and Que on Cytokines Production in the Hippocampus of LPS-Induced Mice

The experimental setup is shown by the treatment scheme (Figure 9A). Hippocampal TNF-α, IL-1β and IL-6 protein levels and blood TNF-α protein levels were significantly increased in the LPS-treated mouse group compared to the control group. The mRNA levels of TNF-α and COX-2 were significantly increased in the hippocampus obtained from LPS-induced mice (Figure 9F,G). Levels of cytokines, such as TNF-α, IL-1β, and IL-6, in the hippocampus were significantly decreased in Ast (or A) and Que (or Q) treated mice compared to those observed in LPS-treated mice (LPS group) (Figure 9B,D,E). In addition, the level of TNF-α in the blood was significantly decreased in the Ast or Que treatment group compared to the LPS treatment group (Figure 9C). The administration of Ast and Que significantly attenuated the increase of TNF-αand COX-2 mRNA by administration of LPS (Figure 9F,G).

### 3.8. Ast and Que Inhibit LPS-Induced Phosphorylation of NF-κB/MAPK Pathwas in the Hippocampus of LPS-Induced Mice

The protective effects of Ast and Que on neuroinflammation through the NF-κB/MAPK pathway in the hippocampus of LPS-induced mice were investigated. LPS highly phosphorylated the NF-κB while co-treatment with LPS plus Ast or Que decreased p-NF-κB level (Figure 10A,C). Additionally, p-Erk, p-p38, and p-JNK were all significantly induced by LPS. The phosphorylation of Erk, p38, and JNK was significantly suppressed by Ast and Que in the hippocampus of mice during LPS stimulation (Figure 10B,D–F).

## 4. Discussion

Microglia are immune cells in the CNS [38]. Neurodegenerative diseases (PD and AD) occur in response to extracellular stimuli, such as the uncontrolled over-activation of microglia in the brain hippocampus. Microglia produce excessive amounts of pro-inflammatory cytokines (TNF-α, IL-1β, and IL-6), ROS, NO, and superoxide (O_2_^−^) [38,39] with activation. Natural compounds can impart neuroprotective effects related to their ability to modulate inflammatory responses associated with neurodegenerative disease [13]. Considering the pathogenesis of LPS-caused neuroinflammation, flavonoid has an effective role in suppressing inflammatory responses in the hippocampus of mice [40].

LPS and cytokine (TNF-α and IL-1β) stimulated microglia cultures to release NO via up-regulation of iNOS and O_2_^−^ via activation of NADPH oxidase complexes, among a host of other anti- and pro-inflammatory factors [41]. Excessive NO secretion *in vivo* causes cytotoxicity and promotes an inflammatory reaction, which mediates the inflammatory response in the CNS and is involved in the development of neurodegenerative diseases. The study showed that Ast and Que significantly decreased NO production by inhibition of iNOS in LPS-activated microglia.

Normally, microglial cells secrete low levels of pro-inflammatory cytokines, but with activation, neuroinflammation is indicated by the expression of increased TNF-α, IL-1β, and IL-6, and eventually progressive neuronal cell death [13]. Patients with PD and AD can present with accumulations of pro-inflammatory cytokines in the serum and brain hippocampus. This presentation is an important indicator of neuronal cell death [42]. In this study, the effects of Ast on TNF-α and IL-1β production in LPS-activated microglia were investigated, and it was demonstrated that Ast significantly inhibited cytokine production.

The subfamilies of MAPKs are Erk, JNK, and p38. The signaling pathway of MAPKs induces the synthesis of inflammatory mediators that induce inflammatory responses. Phosphorylation of MAPK molecules (p-Erk, p-JNK, and p-p38) activates transcriptional regulatory elements, such as NF-kB, to produce NO, ROS, and pro-inflammatory cytokines (TNF-α, IL-1β, and IL-6) [8,43]. In LPS-activated microglia, the Akt signaling pathway plays a crucial role in observed neuroinflammation. Both Ast and Que treatments might mitigate microglial inflammation by modulating the MAPK/Akt pathway. Therefore, an investigation of the effects of Ast and Que on LPS-activated microglia was pursued. Phosphorylation of Akt, p38, Erk, and JNK induced by LPS treatment was decreased by Ast and Que treatments. These results suggest that activation of signaling pathways has a direct impact on the production of inflammatory cytokines.

Increased microglial cell activity by harmful stimuli, such as LPS, leads to neuroinflammation. In cells, excessive ROS can induce pro-inflammatory cytokine and inflammatory mediator expression [44,45]. Therefore, the development of inhibitors of ROS production is important for the treatment and prevention of neurodegenerative diseases. The Ast and Que treatment inhibited LPS-increased ROS generation in microglia, demonstrating their antioxidative role in brain neuroprotection. Ast has high DPPH radical scavenging activity at high concentration but not at low concentration. Que also showed good DPPH radical scavenging activity, higher than that of Ast. Antioxidant enzymes such as HO-1 tightly regulate intracellular ROS levels [45]. The over-expression of HO-1 protects neurons from oxidative stress by virtue of its antioxidant properties [17]. Therefore, the induction of HO-1 has been generally regarded as an adaptive cellular response to the toxicity of oxidative stress [45]. In our study, the increase in HO-1 protein levels induced by LPS stress was further enhanced by Ast (10 μg/mL) and Que (2.5 μg/mL) treatments. Thus, the neuroprotective effects of Ast and Que were ascribed to their antioxidant activity through upregulation of HO-1 expression, as well as the direct removal of intracellular ROS. Ast and Que protect against LPS-induced microglia-mediated cytotoxicity by suppressing TNF-α, IL-1β, and iNOS expression. The suppression is associated with the upregulation of intracellular antioxidant HO-1 and suppression of Akt and MAPK signaling pathways.

Previous animal model studies of PD (LPS-treated mice) showed memory impairment and expression of amyloid β- and p-Tau [40]. LPS induced neuroinflammation and accelerated production of various pro-inflammatory cytokines, including TNF-α, IL-1β, IL-6 and COX-2, via activating NF-*k*B and MAPKs signaling pathways. The Ast or Que significantly reduced LPS-induced production of pro-inflammatory cytokines and blocked the NF-*k*B/MAPKs signaling pathway.

## 5. Conclusions

*In vitro* and *in vivo* studies showed that Ast and Que effectively reduced the expression of pro-inflammatory cytokines, regulating HO-1/MAPK and P13K/Akt signaling pathways. The suppression of inflammatory mediator and reducing both MAPK and P13K/Akt signaling activation, and up-regulation of HO-1 was associated with the neuroprotective effects of Ast and Que in LPS-induced inflammatory models.

## Figures and Tables

**Figure 1 foods-11-01505-f001:**
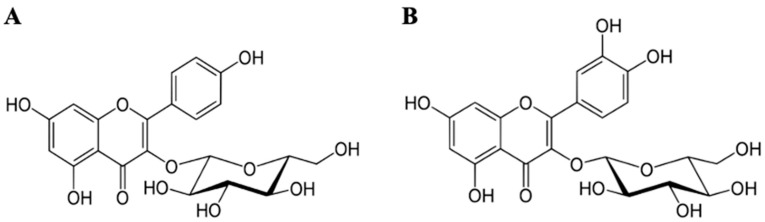
Structures of (**A**) astragalin (Ast) and (**B**) isoquercitrin (Que).

**Figure 2 foods-11-01505-f002:**
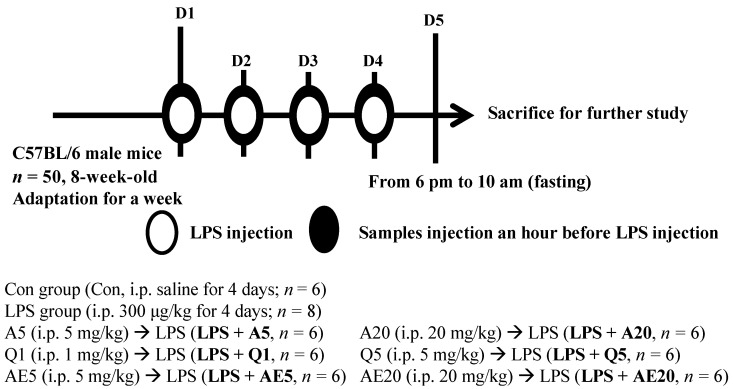
Experiments were carried out according to the schematic illustration. Abbreviations: astragalin (A), isoquercitrin (Q), and 70% ethanol extract of *A. scaber* (AE).

**Figure 3 foods-11-01505-f003:**
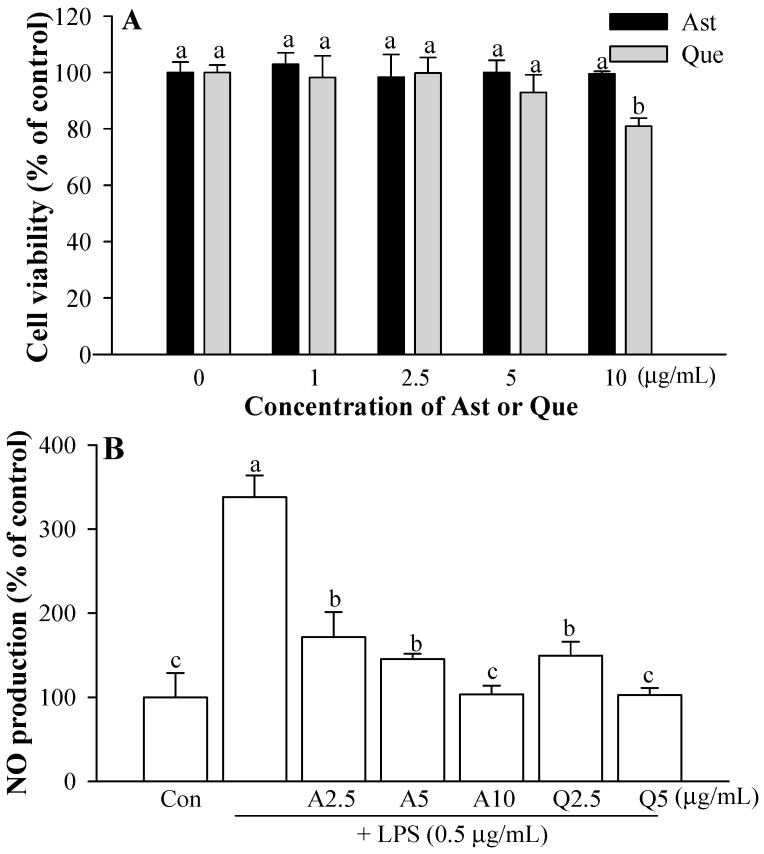
Effect of Ast and Que on (**A**) the cell viability of microglia and (**B**) NO production in LPS-activated microglia. Con, control: LPS- and sample-untreated cells; Group LPS: cell treated with LPS only. The values shown are the mean ± standard deviation (SD) (*n* = 3) and mean labelled with different letters (a–c) within a property differ significantly from each other by Duncan’s multiple range test (*p* < 0.05).

**Figure 4 foods-11-01505-f004:**
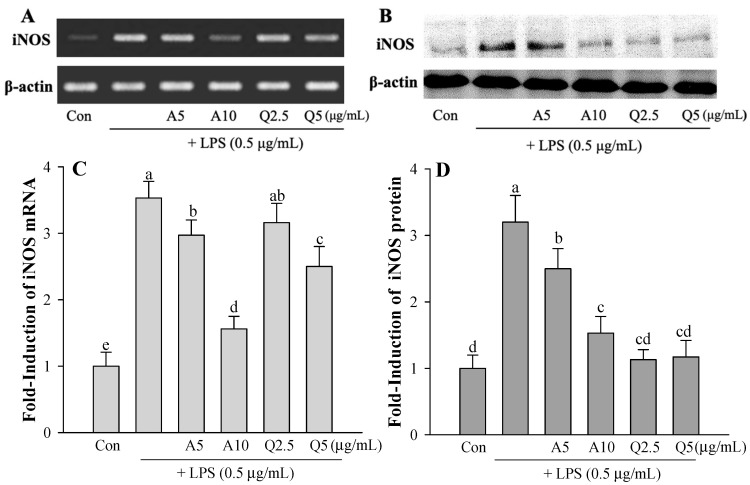
Effect of Ast (A) and Que (Q) on (**A**,**C**) iNOS mRNA and (**B**,**D**) iNOS protein in LPS-activated microglia. Con, control: LPS- and sample-untreated cells; Group LPS: cell treated with LPS only. The level of iNOS mRNA and protein content in each sample were normalized by comparing to the β-actin content. Both mRNA and protein band density were quantified using SigmaGel software. Results are provided as the ratio of each mRNA or protein in treated cells-to-that of the untreated control. The values shown are the mean ± SD (*n* = 3) and means labelled with different letters (a–e) differ significantly as shown by Duncan’s multiple range test (*p* < 0.05).

**Figure 5 foods-11-01505-f005:**
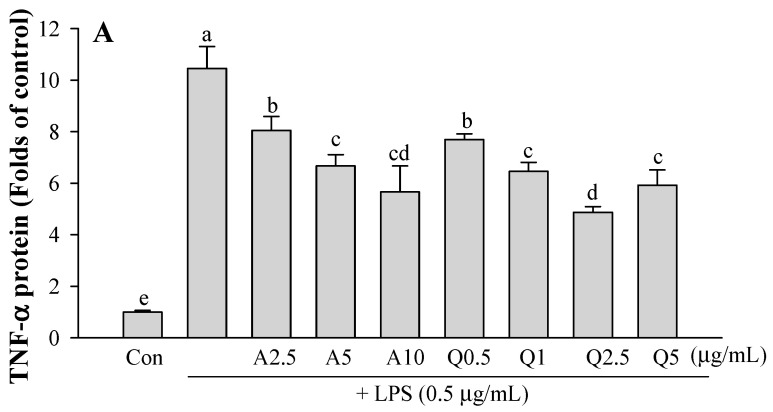
Effect of Ast and Que on TNF-α (**A**) and IL-1β production (**B**) in LPS-activated microglia. Con, control: LPS- and sample-untreated cells; Group LPS: cell treated with LPS only. The values shown are the mean ± SD (*n* = 3) and means labelled with different letters (a–e) differ significantly as shown by Duncan’s multiple range test (*p* < 0.05).

**Figure 6 foods-11-01505-f006:**
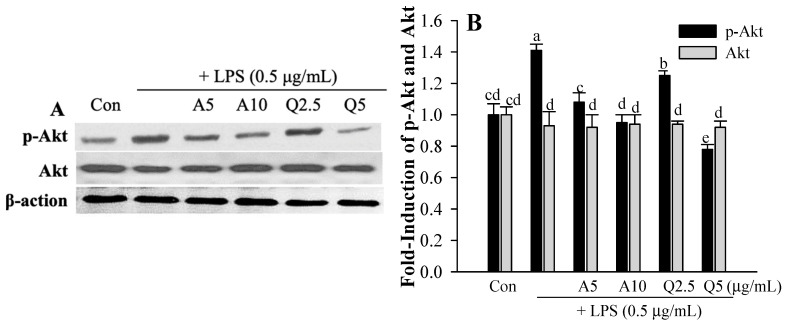
Effect of Ast (or A) and Que (or Q) on LPS-induced phosphorylation of Akt (**A**,**B**) and MAPKs (p38, Erk, and JNK; (**C**–**F**)) pathways in microglia. Con, control: LPS- and sample-untreated cells; Group LPS: cell treated with LPS only. The density of each protein band was quantified by using SigmaGel software. The values shown are the mean ± SD (*n* = 3) and means with different letters (a–e) within a property differ significantly as shown by Duncan’s multiple range test (*p* < 0.05). Akt, protein kinase B; Erk, extracellular protein regulated protein kinase; JNK, c-Jun N-terminal kinase; MAPKs, mitogen-activated protein kinases.

**Figure 7 foods-11-01505-f007:**
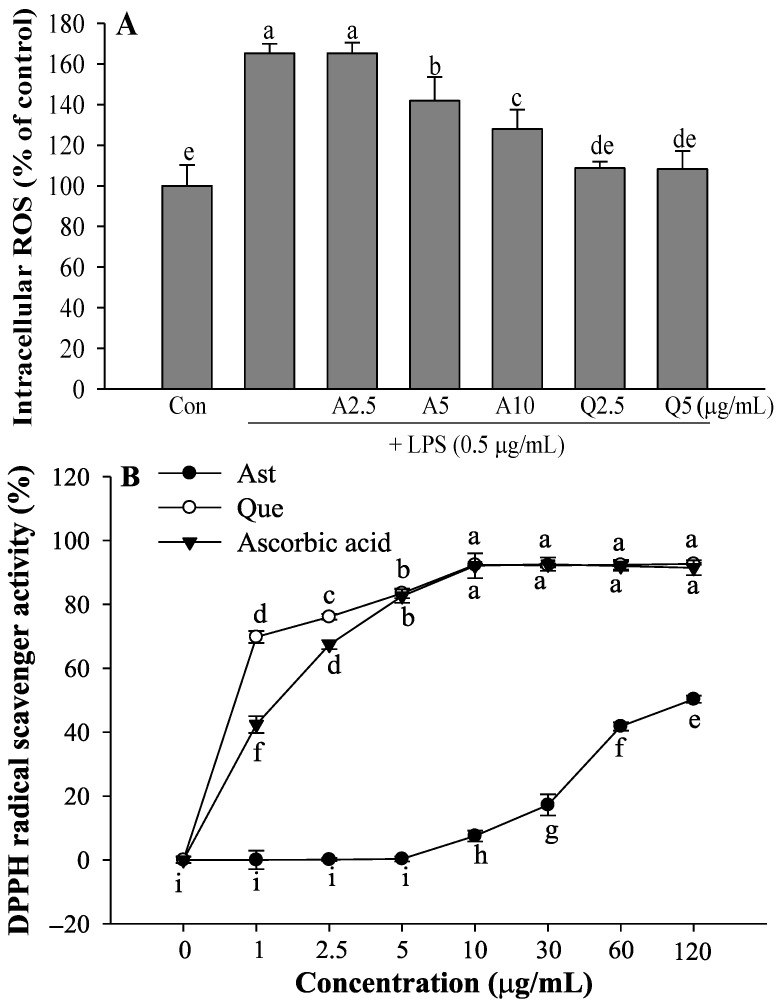
Inhibitory effect of Ast (A) and Que (Q) on LPS-induced intracellular ROS in microglia (**A**), and DPPH radical scavenging activity of Ast and Que (**B**). The values shown are the mean ± SD (*n* = 3) and means labelled with different letters (a–i) within a property differ significantly as determined by Duncan’s multiple range test (*p* < 0.05).

**Figure 8 foods-11-01505-f008:**
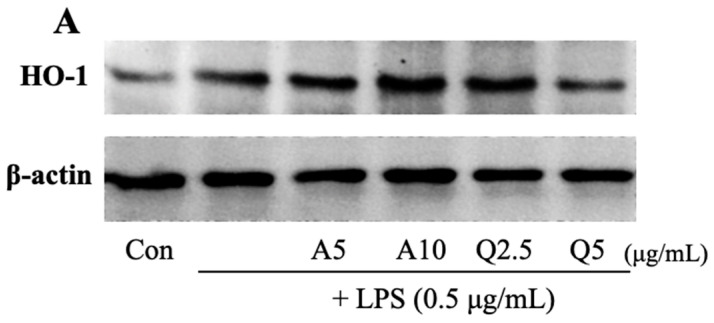
(**A**) Effect of Ast (A) and Que (Q) on HO-1 protein level in LPS-activated microglia. (**B**) For each sample protein content was normalized to the content of β-actin using SigmaGel software. The ratio of the expression of each protein was compared for treated and untreated controls. The values shown are the mean ± SD (*n* = 3) and means labelled with different letters (a–d) within a property differ significantly as determined by Duncan’s multiple range test (*p* < 0.05).

**Figure 9 foods-11-01505-f009:**
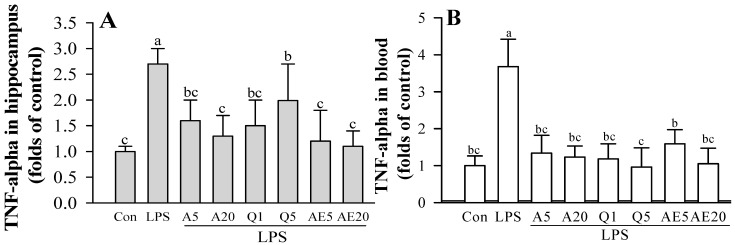
Effect of Ast (A) and Que (Q) on pro-inflammatory cytokines and COX-2 in blood and the hippocampus of LPS induced mice. (**A**,**C**,**D**) protein levels of TNF-α, IL-1β and IL-6 in the hippocampus of LPS-induced mice. (**B**) protein levels of TNF-α in the blood of LPS-induced mice. (**E**–**G**) TNF-α and COX-2 mRNA in the hippocampus of LPS-induced mice. Con, control: LPS- and sample-untreated cells; Group LPS: cell treated with LPS only. Housekeeping gene β-actin level was used for normalization of the mRNA levels in each sample, and the density of each (**E**–**G**) mRNA band was quantified by using SigmaGel software. The values shown are the mean ± SD (*n* = 6) and means labelled with different letters (a–c) within a property differ significantly as determined by Duncan’s multiple range test (*p* < 0.05).

**Figure 10 foods-11-01505-f010:**
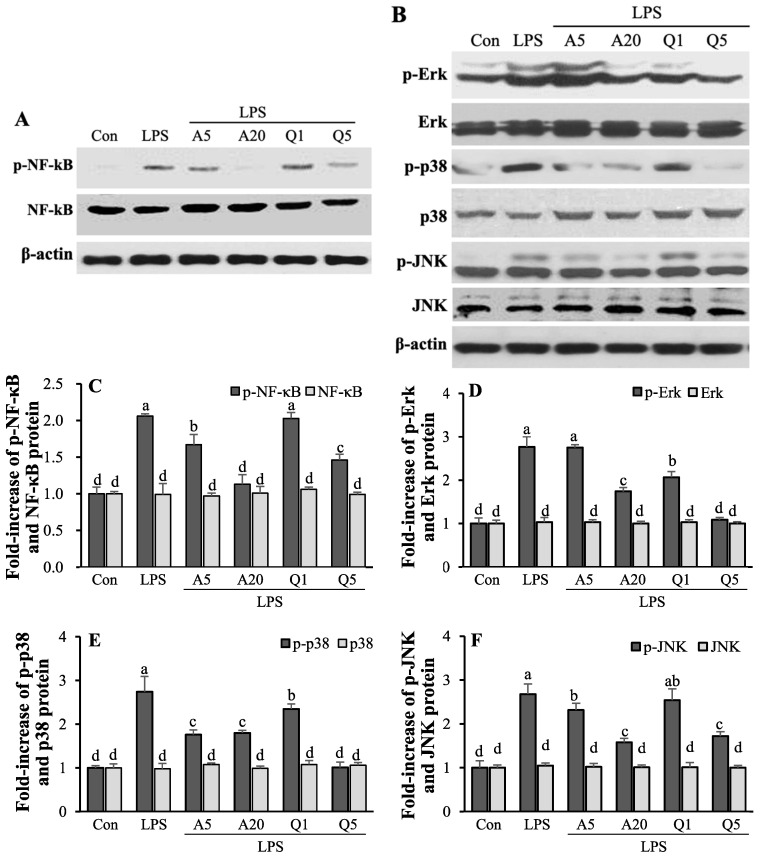
Effects of Ast and Que on the activation of NF-kB, Erk, p38 and JNK in the hippocampus of LPS induced mice (**A**–**F**). The p-NF-*k*B, NF-*k*B, p-Erk, Erk, p-p38, p38, p-JNK, and JNK levels in each sample were normalized to the β-actin level and the density of each protein band was quantified by using SigmaGel software. Con, control: LPS- and sample-untreated cells; Group LPS: cell treated with LPS only. The mRNA levels in each sample were normalized to the data and represent the mean ± SD (*n* = 6). The values shown are the mean ± SD (*n* = 3) and means labelled with different letters (a–d) within a property differ significantly as determined by Duncan’s multiple range test (*p* < 0.05).

## Data Availability

The data of the current study are available from the corresponding author on reasonable request.

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
