# Peer review of "Astragalin and Isoquercitrin Isolated from *Aster scaber* Suppress LPS-Induced Neuroinflammatory Responses in Microglia and Mice"

_foods, 2022, doi:10.3390/foods11101505_

Round 1

Reviewer 1 Report

In this manuscript, the anti-neuroinflammatory effects and mechanism of astragalin (Ast) and isoquercitrin (Que) isolated from chamchwi were studies in lipopolysaccharide (LPS)-activated microglia and hippocampus of LPS induced mice. After establishing cell and mouse models, Ast and Que significantly reduced LPS-induced production of NO, iNOS, and pro-inflammatory cytokines in microglia and hippocampus of mice. Ast or Que inhibited MAPK kinase phosphorylation by extracellular signal-regulated kinase, c-Jun N-terminal kinase, and p38 signaling proteins. Ast and Que inhibited LPS-induced ROS generation in microglia and increased 1,1-diphenyl-2-picrylhydrazyl radical scavenging. In addition, LPS treatment increased heme oxygenase-1 level, which was further elevated after Ast or Que treatments. Ast and Que exert anti-neuroinflammatory activity by down-regulation of MAPKs signaling pathways in LPS-activated microglia and hippocampus of mice. The work is complete and in-depth. The following points should be addressed.

1. In the introduction part, the research progress of two compound activity should be summarized. Have the anti-inflammatory activities of the compounds been studied? Please clarify the purpose and significance of the research and highlight the innovation of the work.

2. Fig. 6A, what does ATK mean?

Author Response

Responses to Reviewer 1 Comments and Suggestions:

In this manuscript, the anti-neuroinflammatory effects and mechanism of astragalin (Ast) and isoquercitrin (Que) isolated from chamchwi were studies in lipopolysaccharide (LPS)-activated microglia and hippocampus of LPS induced mice. After establishing cell and mouse models, Ast and Que significantly reduced LPS-induced production of NO, iNOS, and pro-inflammatory cytokines in microglia and hippocampus of mice. Ast or Que inhibited MAPK kinase phosphorylation by extracellular signal-regulated kinase, c-Jun N-terminal kinase, and p38 signaling proteins. Ast and Que inhibited LPS-induced ROS generation in microglia and increased 1,1-diphenyl-2-picrylhydrazyl radical scavenging. In addition, LPS treatment increased heme oxygenase-1 level, which was further elevated after Ast or Que treatments. Ast and Que exert anti-neuroinflammatory activity by down-regulation of MAPKs signaling pathways in LPS-activated microglia and hippocampus of mice. The work is complete and in-depth. The following points should be addressed. 

  1. In the introduction part, the research progress of two compound activity should be summarized. Have the anti-inflammatory activities of the compounds been studied? Please clarify the purpose and significance of the research and highlight the innovation of the work.

-1 Response: Lines 76-78

The research progress of Ast activity was summarized in the introduction part. The following sentences have been added to the revised manuscript, and relevant references have been inserted in the Introduction and References sections.

Ast down-regulates NF-kB signaling pathway [22] and thereby induces numerous responses including anti-allergic effects [23], cardioprotective effects [24], anti-cancer effects [25], and anti-inflammatory effects [26].

  1. Kim, M.S.; Kim, S.H. Inhibitory effect of astragalin on expression of lipopolysaccharide-induced inflammatory mediators through NF-kB in macrophages. Arch. Pharm. Res. 2011, 34, 2101-2107.
  2. Kotani, M.; Matsumoto, M.; Fujita, A.; Higa, S.; Wang, W.; Suemura, M.; Kishimoto, T.; Tanaka, T. Persimmon leaf extract and astragalin inhibit development of dermatitis and IgE elevation in NC/Nga mice. Journal of Allergy and Clinical Immunology, 2000, 106, 159-166.
  3. Qu, D.; Han, J.; Ren, H.; Yang, W.; Zhang, X.; Zheng, Q.; Wang, D. Cardioprotective effects of astragalin against myocardial ischemia/reperfusion injury in isolated rat heart. Oxid. Med. Cell. Longev. 2016, 2016, 8194690.
  4. Yang, M.; Li, W.Y.; Xie, J.; Wang, Z.L.; Wen, Y.L.; Zhao, C.C.; Tao, L.; Li, L.F.; Tian, Y.; Sheng, J. Astragalin inhibits the proliferation and migration of human colon cancer HCT116 cells by regulating the NF-kB signaling pathway. Front Pharmacol 2021, 12, 639256.
  5. Soromou, L.W.; Chen, N.; Jiang, L.; Huo, M.; Wei, M.; Chu, X.; Martin, F.; Millimouno, F.M.; Feng, H.; Sidime, Y.; Deng, X. Astragalin attenuates lipopolysaccharide-induced inflammatory responses by down-regulating NF-kB signaling pathway. Biochem. Biophys. Res. Commun. 2012, 419, 256-261.

-2 Response: Lines 84-88

The research progress of Que activity was summarized in the introduction part. The following sentences have been added to the revised manuscript, and relevant references have been inserted in the Introduction and References sections.

Que induces anti-inflammatory responses through inhibition of the NF-kB/MAPKs signaling pathway [27,28]. Que bioactivity includes antioxidant effects [29], hepatoprotective effects [30], antiviral activities [31], and a neuroprotective effect on Parkinson’s disease [32]. However, there are no reports describing the anti-neuroinflammatory effect of Ast and Que in vivo and in vitro through down-regulation of the NF-kB/MAPKs signaling pathway.

  1. Rogerio, A.P.; Kanashiro, A.; Fontanari, C.; Silva, E.V.G.; Lucisano-Valim, Y.M.; Soares, E.G.; Faccioli L.H. Anti-inflammatory activity of quercetin and isoquercitrin in experimental murine allergic asthma. Inflammation Research, 2007, 56, 402-408.
  2. Ll L.; Zhang, X.H.; Liu, G.R.; Liu, C.; Dong, Y.M. Isoquercitrin suppresses the expression of histamine and pro-inflammatory cytokines by inhibiting the activation of MAP kinases and NF-kB in human KU812 cells. Chinese Journal of Natural Medicines, 2016, 14, 407-412.
  3. Jung, S.H.; Kim, B.J.; Lee, E.H.; Osborne, N.N. Isoquercitrin is the most effective antioxidant in the plant Thuja orientalis and able to counteract oxidative-induced damage to a transformed cell line (RGC-5 cells). Neurochemistry international, 2010, 57, 713-721.
  4. Xie, W.; Wang, M.; Chen, C.; Zhang, X.; Melzig, M.F. Hepatoprotective effect of isoquercitrin against acetaminophen-induced liver injury. Life Sciences, 2016, 152, 180-189.
  5. Kim, C.H.; Kim, J.E.; Song, Y.J. Antiviral activities of quercetin and isoquercitrin against human herpesviruses. Molecules, 2020, 25, 2379.
  6. Liu, C.; Wang, W.; Li, H.; Liu, J.; Zhang, P.; Cheng, Y.; Qin, X.; Hu, Y.; Wei, Y. The neuroprotective effects of isoquercitrin purified from apple pomace by high-speed countercurrent chromatography in the MPTP acute mouse model of Parkinson’s disease. Food Func. 2021, 12, 6091-6101.

Response: Lines 82-84

The anti-inflammatory activities of astragalin and isoquercitrin have been studied. The anti-inflammatory effects of astragalin and isoquercitrin and those references were added to the introduction and references sections of the manuscript.

Response: Lines 89,90, 93,94

We clarified the purpose and significance of the research. The following sentences have been added to the revised manuscript, and relevant references have been inserted in the Introduction and References sections.

From both in vitro microglia and in vivo animal models, it has been shown that LPS induces neuroinflammation by increasing inflammatory mediators [33].

Therefore, we investigated Ast and Que as functional materials for improving neurodegenerative disorders such as AD-related to neuroinflammation.

  1. Andy, S.N.; Pandy, V.; Alias, Z.; Kadir, H.A. Deoxyelephantopin ameliorates lipopolysaccharides (LPS)-induced memory impairments in rats: Evidence for its anti-neuroinflammatory properties. Life sciences, 2018, 206, 45-60.

Response: Lines 86-88

We highlighted the innovation of the work. We added the following sentences.

However, there are no reports describing the anti-neuroinflammatory effect of Ast and Que in vivo and in vitro through down-regulation of the NF-kB/MAPKs signaling pathway.

  1. Fig. 6A, what does ATK mean?

Response: Fig. 6A, the legend of Fig. 6A, Line 325

Atk mean protein kinase B. Akt was described as the full name at the beginning of the text, and only the abbreviation was indicated afterwards. In addition, Akt was described as the full name in the legend of Fig. 6A. p-Atk and Atk were changed as p-Akt and Akt in Fig. 6A.

Reviewer 2 Report

  • Minor language editing represented by grammar mistakes
  • Line 114 “Cell Viability and NO Ccontent”
  • Animal treatment L138- 144 is vague and the doses with concentrations need to be clearly presented.
  • What are the basis of choice of the doses of Que and Ast in vitro or in vivo
  • Line 162 “primers [20 μL”
  • Line 284-285 “In LPS-activated microglial cells, MAPK and Akt expression both affect the release 284 of pro-inflammatory cytokines [10,20]” no references or citations should be mentioned in results section
  • Line 274-275 “Therefore, Ast and Que may alleviate neurodegenerative diseases through their 294 anti-neuroinflammatory effects.” Explanations or attributions should be avoided in the results section
  • Discission needs mor illustrations of the mechanistic approaches of Que and Ast.
  • Conclusion should be rewritten

Author Response

Responses to Reviewer 2 Comments and Suggestions:

Minor language editing represented by grammar mistakes.

Response: Edited and marked with yellow highlights.

Line 114 “Cell Viability and NO Ccontent”

Response: Line 135

Corrected

Animal treatment L138-144 is vague and the doses with concentrations need to be clearly presented.

Response: Lines 161-170

We clearly presented the doses with a concentration in animal treatment. We added the following sentences (colored sentences).

1) Control group; 2) LPS group [intraperitioneal (i.p.) 300 μg/kg body weight (bw) = 7.5 μg LPS in 0.2 mL/mouse (25 g)]; 3) A5 (i.p. 5 mg/kg bw = 125 μg Ast in 0.2 mL/mouse)  LPS (i.p. 300 μg/kg); 4) A20 (i.p. 20 mg/kg bw = 500 μg Ast in 0.2 mL/mouse)  LPS (i.p. 300 μg/kg); 5) Q1 (i.p. 1 mg/kg bw = 25 μg Que in 0.2 mL/mouse)  LPS (i.p. 300 μg/kg); 6) Q5 (i.p. 5 mg/kg bw = 125 μg Que in 0.2 mL/mouse)  LPS (i.p. 300 μg/kg); 7) AE5 (i.p. 5 mg/kg bw = 125 μg AE in 0.2 mL/mouse)  LPS (i.p. 300 μg/kg); and 8) AE20 (i.p. 20 mg/kg bw = 500 μg AE in 0.2 mL/mouse)  LPS (i.p. 300 μg/kg). Inflammatory progression was induced in each mouse by i.p. injection of LPS (5 mg/kg) for 4 days. The Ast, Que or AE were injected intraperitoneally every day before i.p. injection of LPS (i.p. 300 μg/kg). LPS was dissolved in DPBS. Ast, Que, and AE were dissolved in DMSO and then diluted with DPBS.

What are the basis of choice of the doses of Que and Ast in vitro or in vivo.

Response: Lines 267-272

The doses of Que and Ast in vitro were chosen by MTT assay results. The doses of Que and Ast in vivo were chosen. In animal experiments, in vivo, the doses of Que and Ast were chosen based on published references (Zhang et al., 2018; Lee et al., 2015) on natural compounds from isolated plants.

Zhang, F.X.; Xu, R.S. Juglanin ameliorates LPS-induced neuroinflammation in animal models of Parkinson’s disease and cell culture via inactivating TLR4/NF-kB pathway, Biomedicine & pharmacotherapy, 2018, 97, 1011-1019.

Lee, H.G.; Kim, D.H.; Kim, Y.S. Effects of kaempferol on lipopolysaccharides-induced inflammation in mouse brain. Kor. J. Herbology 2015, 30, 77-84.

Line 162 “primers [20 μL”

Response: Line 187

Changed to parentheses

Line 284-285 “In LPS-activated microglial cells, MAPK and Akt expression both affect the release of pro-inflammatory cytokines [10,20]” no references or citations should be mentioned in Results section.

Response: Line 316

Deleted

Line 294-295 “Therefore, Ast and Que may alleviate neurodegenerative diseases through their anti-neuroinflammatory effects.” Explanations or attributions should be avoided in the results section.

Response: Line 325

Deleted

Discussion needs more illustrations of the mechanistic approaches of Que and Ast.

Response: Lines 451-454

The explanation of the mechanistic approaches of Que and Ast was added in the Discussion section. The following sentences have been added to the revised manuscript.

Ast and Que protect against LPS-induced microglia-mediated cytotoxicity by suppressing TNF-α, IL-1β, and iNOS expression. The suppression is associated with the upregulation of intracellular antioxidant HO-1 and suppression of Akt and MAPK signaling pathways.

Conclusion should be rewritten.

Response: Lines 462-466

We rewrote the Conclusions. The following sentences have been changed to the revised manuscript.

In vitro and in vivo studies showed that Ast and Que effectively reduced the increased expression of pro-inflammatory cytokines, regulating HO-1/MAPK and P13K/Akt signaling pathways. The suppression of inflammatory mediator and reducing both the MAPK and P13K/Akt signaling activation, and up-regulation of HO-1 was associated with the neuroprotective effects of Ast and Que in LPS-induced inflammatory models.

Reviewer 3 Report

A manuscript entitled “Anti-neuroinflammatory Activities Astragalin and Isoquercitrin 2 Isolated from Aster scaber Extract” which elostrate Ast and Que’s anti-neuroinflammatory effects and action mechanism isolated from A. scaber were investigated in LPS-activated microglia and hippocampus of LPS induced mice.The authors have written very well some how; some minor modifications of the submitted paper are recommended.

In the abstract section objective is not clear so must revise the abstract

Authors may revise the introduction section; also, they can add other properties of Aster scaber.

Identification of material is not mentioned in the manuscript

Grammatical errors are found

Author Response

Responses to Reviewer 3 Comments and Suggestions:

A manuscript entitled “Anti-neuroinflammatory Activities Astragalin and Isoquercitrin 2 Isolated from Aster scaber Extract” which elostrate Ast and Que’s anti-neuroinflammatory effects and action mechanism isolated from A. scaber were investigated in LPS-activated microglia and hippocampus of LPS induced mice. The authors have written very well some how; some minor modifications of the submitted paper are recommended.

In the abstract section objective is not clear so must revise the abstract

Response: Lines 19-21

We revised the Abstract section.

The current study investigated the effect and mechanism of astragalin (Ast) and isoquercitrin (Que) isolated from chamchwi (Aster scaber Thunb.) on anti-neuroinflammatory effects in lipopolysaccharide (LPS)-activated microglia and hippocampus of LPS induced mice.

Authors may revise the introduction section; also, they can add other properties of Aster scaber.

Response:  Lines 73-77

We revised the introduction section. Also, we added other properties of A. scaber Thunb. The following sentences have been added to the revised manuscript, and relevant references have been inserted in the Introduction and References sections.

Chamchwi (Aster scaber Thunb.), an edible plant rich in flavonoids, is widely cultivated as a culinary vegetable in Korea (Chung et al., 2016). The rich nutrients contained in A. scaber are vitamin C, Ca, Fe, and β-carotene (Chung et al., 1993). A. Scaber leaves contain caffeoylquinic acid compounds, flavonoids, and terpenoids (Jung et al., 2001; Choi et al., 2020). A. Scaber had the potential antioxidant and anti-obesity effects (Choi et al., 2013) due to radical scavenging-linked anti-adipogenic activity Choi et al., 2020) and has protective effects against oxidative stress-induced human brain cell death (Chung et al., 2016).

Chung, M.J.; Lee, S.H.; Park, Y.I.; Lee, J.S.; Kwon, K.H. Neuroprotective effects of phytosterols and flavonoids from Cirsium setidens and Aster scaber in human brain neuroblastoma SK-N-SH cells. Life Sci. 2016, 148, 173–182.

Chung, T.Y.; Eiserich, J.P.; Shibamoto, T. Volatile compounds isolated from edible Korean chamchwi (Aster scaber Thunb). J. Agric. Food Chem. 1993, 41, 1693-1697.

Jung, C.M.; Kwon, H.C.; Seo, J.J.; Ohizumi, Y.; Matsunaga, K.; Saito, S.; Lee, K.R. Two new monoterpene peroxide glycoside from Aster scaber. Chem. Pharm. Bull. 2001, 49, 912-914.

Choi, Y.E.; Choi,S.I.; Han, X.; Men, X.; Jang, G.W.; Kwon, H.Y.; Kang, S.R.; Han, J.S.; Lee, O.H. Radical scavenging-linked anti-adipogenic activity of Aster scaber ethanolic extract and its bioactive compound. Antioxidants, 2020, 9, 1290.

Choi, J.H.; Park, Y.H.; Lee, I.S.; Lee, S.P. Antioxidant activity and inhibitory effect of Aster scaber Thunb. Extract on adipocyte differentiation in 3T3-L1 cells. Korean J. Food Sci. Technol. 2013, 45, 356-363.

Identification of material is not mentioned in the manuscript

Response: lines 98-100

Identification of Aster scaber is mentioned in the manuscript. The following sentences have been added to the revised manuscript.

The specimen was authenticated by Prof. Seon Haeng Cho, Gongju National University of Education (Chungcheongnam-do, Korea).

Grammatical errors are found

Response: Corrected

Round 2

Reviewer 2 Report

THANK YOU FOR APPROPRIATE RESPONSE